# RNA-Seq Analysis Reveals an Essential Role of the cGMP-PKG-MAPK Pathways in Retinal Degeneration Caused by Cep250 Deficiency

**DOI:** 10.3390/ijms24108843

**Published:** 2023-05-16

**Authors:** Chong Chen, Yu Rong, Youyuan Zhuang, Cheng Tang, Qian Liu, Peng Lin, Dandan Li, Xinyi Zhao, Fan Lu, Jia Qu, Xinting Liu

**Affiliations:** 1National Engineering Research Center of Ophthalmology and Optometry, Eye Hospital, Wenzhou Medical University, Wenzhou 325027, China; chenchong@eye.ac.cn (C.C.); yvierong@163.com (Y.R.); zhuangyy@wmu.edu.cn (Y.Z.); c249325687@163.com (C.T.); liuxqian2420@163.com (Q.L.); pplinpeng@163.com (P.L.); lddwasei@163.com (D.L.); zxyoptometry@163.com (X.Z.); lufan62@mail.eye.ac.cn (F.L.); 2State Key Laboratory of Ophthalmology, Optometry and Visual Science, Eye Hospital, Wenzhou Medical University, Wenzhou 325027, China

**Keywords:** Usher syndrome, retinal degeneration, Cep250-KO mice, RNA sequencing, cGMP-PKG-MAPK signalling pathways

## Abstract

Usher syndrome (USH) is characterised by degenerative vision loss known as retinitis pigmentosa (RP), sensorineural hearing loss, and vestibular dysfunction. RP can cause degeneration and the loss of rod and cone photoreceptors, leading to structural and functional changes in the retina. Cep250 is a candidate gene for atypical Usher syndrome, and this study describes the development of a Cep250 KO mouse model to investigate its pathogenesis. OCT and ERG were applied in Cep250 and WT mice at P90 and P180 to access the general structure and function of the retina. After recording the ERG responses and OCT images at P90 and P180, the cone and rod photoreceptors were visualised using an immunofluorescent stain. TUNEL assays were applied to observe the apoptosis in Cep250 and WT mice retinas. The total RNA was extracted from the retinas and executed for RNA sequencing at P90. Compared with WT mice, the thickness of the ONL, IS/OS, and whole retina of Cep250 mice was significantly reduced. The a-wave and b-wave amplitude of Cep250 mice in scotopic and photopic ERG were lower, especially the a-wave. According to the immunostaining and TUNEL stain results, the photoreceptors in the Cep250 retinas were also reduced. An RNA-seq analysis showed that 149 genes were upregulated and another 149 genes were downregulated in Cep250 KO retinas compared with WT mice retinas. A KEGG enrichment analysis indicated that cGMP-PKG signalling pathways, MAPK signalling pathways, edn2-fgf2 axis pathways, and thyroid hormone synthesis were upregulated, whereas protein processing in the endoplasmic reticulum was downregulated in Cep250 KO eyes. Cep250 KO mice experience a late-stage retinal degeneration that manifests as the atypical USH phenotype. The dysregulation of the cGMP-PKG-MAPK pathways may contribute to the pathogenesis of cilia-related retinal degeneration.

## 1. Introduction

Inherited retinal diseases (IRDs) that are caused by nuclear and mitochondrial gene variants are complex and heterogeneous, damaging different parts of the retina and leading to visual impairment [1]. Out of all IRDs, retinitis pigmentosa (RP) is the most common globally, accounting for nearly 1 out of 4000 cases. RP is characterised as the degeneration and loss of rod and cone photoreceptors, which manifests as an attenuation of the outer nuclear layer (ONL), which consists of the nucleus of rod and cone photoreceptors [2]. In contrast, the inner nuclear layer (INL), containing amacrine cells, bipolar cells, and horizontal cell neurons, is relatively well preserved in the early stage of RP. The typical clinical presentation of patients with retinitis pigmentosa is a gradual loss of night vision, peripheral vision, and central vision, which results from the progressive loss of photoreceptors [3]. RP is classified as syndromic or non-syndromic, with the syndromic type accounting for 20–30% of cases. These cases are associated with non-ocular disease and fall within over 30 various syndromes, such as Usher’s syndrome and Bardet–Biedl syndrome, impacting retinal pigment epithelium cell function, photoelectric transformation pathways, mRNA shear factors, and ciliary formation [3].

Usher syndrome (USH), characterised by the combination of degenerative vision loss known as retinitis pigmentosa (RP), sensorineural hearing loss (SNHL), and sometimes vestibular dysfunction is a rare disease [4]. It was traditionally classified into three subtypes (USH1, USH2, and USH3) according to the age of onset, severity, symptom progression, and vestibular dysfunction. USH1, USH2, and USH3 are the most severe, common, and relatively rare forms of the syndrome, respectively [5]. USH1 patients have severe-to-profound bilateral sensorineural hearing loss (SNHL) and RP at an early age. USH2 patients have moderate-to-severe congenital HL and a late onset of RP, and these symptoms are not as severe as USH1 patients [6,7]. Other than the three subtypes, patients who do not match symptoms in these three cohorts are classified as atypical USH [5]. Until now, ten genes have been correlated with typical USH (MYO7A, USH1C, CDH23, PCDH15, USH1G, CIB2, USH2A, ADGRV1, WHRN, CLRN1) [7]. More genes related to atypical USH and their mechanisms remain to be understood.

The Cep250 gene encodes the C-Nap1 protein, an essential member of the centrosome activating element CEP family and a critical protein that regulates the adhesion of centrosomes during intermitosis [8,9]. Two members, CEP290 and CEP164, have been shown to have mutations that cause symptoms of RP [10,11]. Previous studies report that the C-Nap1 protein plays an important role in cell cycle regulation, is localised in the basal body of retinal photoreceptor cilia, and interacts with several ciliary proteins that are closely related to retinal function, such as rootletin and NEK2 [12,13]. CEP250 was first revealed in an Iranian Jewish family with USH through whole exome sequencing (WES), revealing two nonsense mutations, c.3289C > T (p.Q1097 *) in C2orf71 and c.3463C > T (p.R1155 *) in CEP250 [14]. Two years later, a cohort of 33 pedigrees with various retinal disorders was analysed using the WES technique, identifying another mutant in CEP250 (Cep250 c.1826C > T p.A609V) that caused non-syndromic IRD and influenced cilia formation [15]. A later study in a Japanese family verified that compound heterozygous variants c.361C > T and c.562C > T in CEP250 were associated with mild cone–rod dystrophy and sensorineural hearing loss [16]. In 2019, Huang et al reported a mutation (c.562C > T, p.R188 *) in the CEP250 in a consanguineous family with non-syndromic RP and constructed the homozygous knock-in mice model, which showed significantly reduced retinal thickness [17]. Previous studies support the hypothesis that CEP250 may be a candidate gene for atypical USH. However, the essential role of Cep250 mutants leading to an atypical USH phenotype is unclear. 

Although the mechanisms of IRDs or RP have been extensively studied in human and animal models, there are scarce reports on the biomolecular pathways involved in cilia-induced retinal degeneration. To investigate the functional impact of Cep250, a late-stage retinal degeneration using CRISPR-Cas9 knockout mice was used, which confirmed that the KO eyes experienced degeneration and loss in rod and cone photoreceptors. Further, RNA sequencing analysis uncovered the essential role of the cGMP-PKG-MAPK pathways in cilia-associated retinal degeneration disorder.

## 2. Results

### 2.1. Retinal Structure Changes Cep250 Mice

Spectral-domain optical coherence tomography (SD-OCT) was applied to Cep250 and WT mice at P90 and P180 to analyse the thickness and structural changes in the retina (Figure 1A). We observe that different retina layers showed varying thickness changes between Cep250 mice and WT mice. Compared with WT mice, the ONL thickness of Cep250 mice was significantly reduced (Figure 1B). As ONL mainly consists of the nuclei of cone and rod cells, the photoreceptors of Cep250 mice were somewhat damaged. Notably, the thickness of the combined inner and outer segment layers (IS/OS) also manifested differences between the two groups. The Cep250 mice had thinner retinal layers, with the thickness of the whole retina showing the same trend (Figure 1C,D). The same trend was found at P180 (Figure 1E–G).

### 2.2. Impairment of Retina Function in Cep250 Mice

The ERG responses of Cep250 and WT mice were recorded at P90 and P180 to reflect their retinal function (Figure 2A). Differences were observed between Cep250 and WT mice at diverse light stimulation. Notably, the a-wave amplitude of Cep250 mice in scotopic ERG (Dark 3.0 and 10.0) was reduced (Figure 2B). In contrast, the same trend was not observed in the phototic responses, with no significant differences between the two groups under this condition (Figure 2C). Regarding the b-wave amplitude, although WT mice had relatively higher scotopic ERG levels, the two mice groups did not exhibit significant statistical differences; there was a smaller gap in the photopic responses (Figure 2D,E). The same trend was also confirmed at P180 (Appendix A). Even so, the results have shown the impairment of retina function in Cep250 mice.

### 2.3. Reduction of Cone and Rod Photoreceptors in Cep250 Retinas

After recording the ERG responses and OCT images at P90 and P180, immunofluorescence staining of the cone and rod photoreceptors was used to deduce the number of photoreceptors to access the retina function. Rhodopsin, M-opsin, and S-opsin antibodies were applied in this experiment. The results revealed that the retina of Cep250 mice had a smaller proportion of rhodopsin proteins than WT mice (Figure 3A). The same trend was also observed following M-opsin and S-opsin immunostaining (Figure 3B,C). In addition to these OS protein changes, the immunostaining pages highlighted the ONL thickness that was stained using DAPI. The ONL of Cep250 mice were apparently thinner than WT mice. These results further indicate the structural impairment and reduction of cone and rod photoreceptors of Cep250 mice. TUNEL staining showed that Cep250 KO mice had more apoptosis in the ONL layer (Figure 3D). 

### 2.4. RNA-Sequencing Analysis of Molecular Changes in Cep250 Mice Retinas

In the RNA-seq, a fold change (FC) of at least two and a *p*-value < 0.05 were the criteria for defining differentially expressed genes (DEGs). In total, 298 genes were differentially expressed in the Cep250 KO group, of which 50% were noted to be upregulated (Figure 4A). The top 15 upregulated genes and downregulated genes are shown in Table 1. A Gene Ontology (GO) enrichment analysis revealed that the DEGs in the Cep250 KO group are primarily involved in biological processes (such as regulation of ion transmembrane transport and signal transduction), cellular components (such as membrane region and neuronal cell body), and molecular functions (such as calcium ion and metal ion binding) (Appendix A). KEGG pathway enrichment analysis indicated that the DEGs are primarily involved the cGMP-PKG signalling pathway, MAPK signalling pathway, insulin secretion, edn2-fgf2 axis pathways, and thyroid hormone synthesis (Figure 4B). 

Eight genes involved in the cGMP-PKG signalling pathway (Pde3a, Pde3b, Pde6b), MAPK signalling pathway (Parp1, Fgf2), cAMP signalling pathways (Edn2), JAK-STAT signaling pathway (Gfap), and in tyrosine metabolism (Tyr) were selected and validated using qRT-PCR. The results show that compared with the WT group, the expression levels of Pde3a, Pde3b, Gfap, Edn2, Fgf2, and Tyr were significantly increased in the Cep250 KO cohort and showed similar changes in the RNA-seq and qPCR analyses. In contrast, the Pde6b expression levels significantly decrease in the Cep250 KO group but showed opposite alterations in RNA-seq (up) and qPCR (down). Finally, we confirmed Parp1 by qPCR and showed an increase in the Cep250 KO cohort compared with the WT group (Figure 4C).

## 3. Discussion

This study designed and produced Cep250 knockout mice in a C57BL/6J background to access their retinal abnormalities, and we investigated the essential role of cilia-related gene mutants leading to the atypical USH phenotype. The OCT results indicate that the thickness of the ONL, IS/OS layer, and whole retina were significantly reduced in Cep250 knockout mice, leading to the reduction of photoreceptor cells. Over time, this trend becomes more remarkable. Additionally, immunostaining illustrated that the cone, S-opsin, and M-opsin photoreceptors were diminishing. ERG also suggests the loss of retinal function in Cep250 knockout mice to some extent, although not markedly. Abu-Diab et al. describe another homozygous knockout of Cep250 with a relatively late onset of retinal degeneration in which P180 did not reveal a reduced ONL thickness and ERG response; meanwhile, in P360, these parameters were diminished [8]. We suppose that the two reasons below may account for the differences between our studies. First, in the above-mentioned manuscript, the Cep250 KO mouse model was generated by activating a construct harbouring a deletion of exons 6 and 7. In our mouse model, 8593 bp sequences from exon 3 to exon 12 were removed, which may result in earlier and more typical functional and structural degeneration. Second, the thickness of ONL in that manuscript were measured through the images of mid-peripheral retina stained with H&E. In our study, we took OCT images and then calculated the thickness of each layer using MATLAB software. In general, although the typical changes of retinal structure and function in our mice models appeared at different time nodes, they manifested the same trend of retinal degeneration.

RNA-seq technology demonstrated that many differentially expressed genes in the Cep250 KO mice retina were enriched in several upregulated pathways, including the cGMP-PKG signalling pathways, cAMP signalling pathways, and MAPK signalling pathways. cGMP signalling is critical in photoreceptor cell death caused by disease-causing mutations. Studies suggest the hypothesis that high intracellular cGMP levels trigger photoreceptors’ degeneration [18]; examples of such a situation are genes that encode for photoreceptor phosphodiesterase-6 (PDE6) [19]. Among all PDE6 family genes, Pde6a, Pde6b, and Pde6g can cause RP [20,21,22]. cGMP-dependent protein kinase (PKG) is a key effector of cGMP-signalling, and its overactivation may trigger photoreceptor cell death [23]. The underlying mechanism could be that the disease-causing mutations in the PDE6B induce photoreceptor cGMP accumulation. This, in turn, activates parallel activation of protein kinase G (PKG) and MAPK that have involved two core molecules, Parp1 and Egr1 [24].

Previous studies have shown that leukaemia inhibitory factor (LIF) can be part of a retinal defence mechanism that increases the survival of visual cells when suffering light damage [25]. Damaged photoreceptors can release endothelin-2 (Edn2) to induce Müller cells to release fibroblast growth factor-2 (Fgf2), which supports the survival of viable photoreceptors [26]. In the rd10 mouse retina, an RP mouse model, protective genes such as LIF, fgf2, and edn2 are also extensively upregulated [27]. Norrin, a secreted protein that protects retinal ganglion cells (RGCs) via LIF, is required to enhance Müller cell gliosis and induce protective factors such as Gfap, Edn2, and Fgf2 [28]. Such data lead to the hypothesis that fgf2 and edn2 are protective genes in Cep250 KO mice retina. 

Besides the two pathways mentioned above, several genes were also shown to be significantly upregulated or downregulated that are closely related to retinal function. Gene expression of Tyr and Serping1 were upregulated, whereas Nnat and crystallin-related genes (Cryaa, Crygs, Cryba1, Crybb1, Cryab) were downregulated in Cep250 mice. Nnat can be detected in embryonic and adult neuroretina and corneal epithelial, inferring that Nnat may be essential during eye development [29]. Nnat is primarily expressed in rods in mammals and co-localises with rhodopsin. Nnat can be more highly expressed in stressed retinas, suggesting that Nnat is a novel stress-responsive protein with a potential structural and/or functional role in adult mammalian retinas [30]. Interestingly, in a partial transection (PT) of the optic nerve, compared with an uninjured retina, Crygs, Cryba1, and Cryba2 were significantly downregulated in an injured dorsal retina at day 1 and 7 [31]. These results demonstrated that crystallin-related genes dynamically interact between neuroprotective and neurodegenerative events. 

Tyr encodes tyrosinase, the first and rate-limiting enzyme in biosynthesis. Mutations in Tyr may cause loss of pigment production, leading to a rare genetic disease: albinism. Tyr is expressed in two cell types in mice: neural crest-derived melanocytes and the retinal pigment epithelium (RPE) [32]. A previous study showed that the tyrosine metabolic pathway is dysregulated in myopic retinal degenerated eyes [33]. The potential relationship between Tyr and retinal degeneration remains unknown. Serping1, encoding C1INH, is a glycoprotein that can inhibit complement activation. It exists in the human retina and RPE/choroid [34]. A previous study reported that more C1INH is expressed in the choroid of AMD eyes compared with normal eyes [35]. Serping1 participated in an innate immune system that could be activated by optic nerve crush. Among the genes in this endogenous immune system, many genes are risk factors for AMD [36].

## 4. Materials and Methods

### 4.1. Animals

The Cep250 (ENSMUSG00000038241) knockout model used in present study was shared by Dr. Xiufeng Huang (The Second Affiliated Hospital of Wenzhou Medical University). The CRISPR/Cas9 system was used to generate Cep250 knockout mice in C57BL/6J background animals. The Cep250 gene of a mouse is located on chromosome 2. In this study, sgRNAs targeting introns 2 and 12 were applied (8593 bp sequences from exon 3 to exon 12 were removed). Founders were genotyped using DNA sequencing and a T7E1 analysis. The genotyping primers, design strategy, example genotyping results, and qRT-PCR analysis results are shown in Appendix A. In this study, we define C57BL/6J and Cep250 mice as wild-type and mutant mice, respectively.

### 4.2. Aminal Care

The mice were raised in the School of Ophthalmology and Optometry of Wenzhou Medical University under a standard 12-h light/dark cycle and were provided with a normal diet and tap water. All experiments followed the ARVO (Association for Research in Vision and Ophthalmology) statement for the Use of Animals and are approved by the Institutional Animal Care and Use Committee of Wenzhou Medical University.

### 4.3. Optical Coherence Tomography Image Acquisition and Analysis

A Micron IV retinal imaging system (Pleasanton, CA, USA) was used to acquire fundus and retinal layer images of Cep250 and WT mice. Firstly, 0.5% tropicamide ophthalmic drops were used to dilate their pupils for at least 5 min. Then, the animals were anaesthetised using intraperitoneal injection with ketamine (80 mg/kg) and xylazine (16 mg/kg). Ofloxacin eye ointment (Dicolol, Shenyang, China) was applied to the eyes to keep the cornea moist. The mice were then placed to face the camera of the micron Ⅳ, and retinal OCT images were acquired through the centre of the optic nerve (ON). The ImageJ software program (version 1.53, National Institutes of Health, Bethesda, MD, USA) was used to manually define the different layers of the retinas. The thicknesses of the different layers (ONL, IS/OS, and the whole retina) were calculated using MATLAB (The Mathworks Inc., Natick, MA, USA) software.

### 4.4. Electroretinogram

Scotopic and photopic responses in both eyes of an individual mouse were recorded using a stable Ganzfeld dome stimulating and data collecting system (Q450SC UV; Roland, Wiesbaden, Germany). Mice were dark-adapted for over 12 h and anaesthetised with ketamine and xylazine under dim red light. Their pupils were dilated with 0.5% tropicamide ophthalmic drops for at least 5 min. Drops of 2.5% methylcellulose were applied to the cornea to improve conjunction with a corneal gold wire electrode. The mice were then positioned on a heated platform at 37 °C to maintain their body temperature and avoid opacification of the refractive interstitium. Ground electrodes and a referential needle were punctured into the tail and cheek. During the experiment, 0.9% sodium chloride was applied to keep the cornea moist. The applied ERG parameters for photopic responses were 0.48 log candela (cd)·s/m^2^ (light 3.0). The ERG parameters for scotopic responses were 2.02 log cd·s/m^2^ (dark 0.01), 0.48 log cd·s/m^2^ (dark 3.0), and 0.98 log cd·s/m^2^ (dark 10.0).

### 4.5. Immunohistochemistry and TUNEL Staining

Eyeballs were extracted from euthanised mice and fixed in 4% paraformaldehyde for 1 h. Then, the whole eyeballs were subjected to 30% sucrose overnight (4 °C) to dehydrate them. Ultimately, they were embedded in an optimal cutting temperature compound (Tissue Tek, Torrance, CA, USA) and cryo-sectioned longitudinally at a thickness of 10 μm. Sections were permeated with 0.1% Triton X-100 and blocked in 5% bovine serum albumin (BSA) at room temperature. Then, they were incubated overnight at 4 °C with one of the following antibodies: rhodopsin antibody (clone B6-30, cat# NBP2-25160, Novus Biologicals, Centennial, CO, USA, 1:1000); opsin 1 antibody (Novus Biologicals cat# 110–74730, Novus Biologicals, Littleton, CO, USA, 1:1000); or S-opsin antibody (Novus Cat# NBP1–20194, Novus Biologicals, Littleton, CO, USA, 1:100). The sections were then incubated with corresponding secondary antibodies for 1 h at room temperature. The secondary antibodies were goat anti-rabbit IgG H&L (Alexa Fluor^®^ 488) (1:500, Abcam, ab150077) and goat anti-mouse IgG H&L (Alexa Fluor^®^ 488) (Abcam, ab150113; 1:1000). Ultimately, the samples were stained with 4′,6-diamidino-2-phenylindole (DAPI). A TUNEL analysis was performed using the One Step TUNEL Apoptosis Assay Kit (Beyotime Biotechnology, Haimen, China) based on the kit’s instructions. The images were obtained using a DM4B (Zeiss, Oberkochen, Germany) microscope.

### 4.6. RNA Sequencing

RNA Extraction: Three eye retinas from the Cep250 KO and WT groups were used for the RNA sequencing experiments. The total RNA was extracted using the Trizol reagent kit (Invitrogen, Carlsbad, CA, USA) according to the manufacturer’s instructions. The RNA quantity was measured using Qubit 2.0 and Nanodrop One (Thermo Fisher Scientific, Waltham, MA, USA), and RNA integrity was determined using an Agilent 2100 Bioanalyzer (Agilent Technologies, Palo Alto, CA, USA). The RNA sequencing was performed by Biomarker Biotechnology Co. (Beijing, China) using the HiSeq3000 Sequencing System (Illumina, San Diego, CA, USA). The quality of the sequencing data was assessed using FASTQC 0.18.0. The mRNA sequences were mapped to the genome (GRCm38) using HISAT 2.2.4. The counts of each gene were extracted from the mapping files using StringTie 1.3.3. The RNA differential expression in different groups was analysed using DESeq2 software. WebGestalt (http://www.webgestalt.org/ (accessed on 2 July 2019) and g:Profiler (http://biit.cs.ut.ee/gprofiler/gost (accessed on 20 October 2020) were used to generate Gene Ontology (GO) terms and Kyoto Encyclopedia of Genes and Genomes (KEGG) pathways [37,38].

### 4.7. Quantitative Real-Time PCR (qRT-PCR)

Neural retinas were isolated, and the total RNA was extracted using a total RNA extraction kit (Vazyme, Nanjing, China). Reverse transcription was performed using a reverse transcription kit (Vazyme, Nanjing, China). The qRT-PCR analysis was conducted using ABI-Q6. The primers used in qRT-PCR are shown in Appendix A. The relative expression of candidate genes was obtained through the comparative threshold cycle (2^−∆∆Ct^) method.

### 4.8. Data Analysis

The data are shown as the means ± SEM; OCT and ERG data were analysed using Student’s *t*-test between the two groups. The results were visualised using GraphPad Prism software. * *p* < 0.05, ** *p* < 0.01, *** *p* < 0.001, and **** *p* < 0.0001 were used when comparing the Cep250 and C57 control groups.

## 5. Conclusions

This study used Cep250 knockout mice in a C57BL/6J background to uncover its specific phenotype and mechanism. Compared with WT mice, the thickness of the ONL, IS/OS, and whole retina of Cep250 mice was significantly reduced. Meanwhile, the a-wave and b-wave amplitude of Cep250 mice in scotopic and photopic ERG were lower, particularly in the a-wave. According to the immunostaining results, the number of photoreceptors in Cep250 retinas was reduced. Mechanistically, the cGMP-PKG signalling pathway is dysregulated and the edn2-fgf2 axis pathways are activated in Cep250 KO eyes. The causal link between cGMP-PKG-MAPK and cilia-related retinal degeneration is novel and exciting, and this link warrants further investigation.

## Figures and Tables

**Figure 1 ijms-24-08843-f001:**
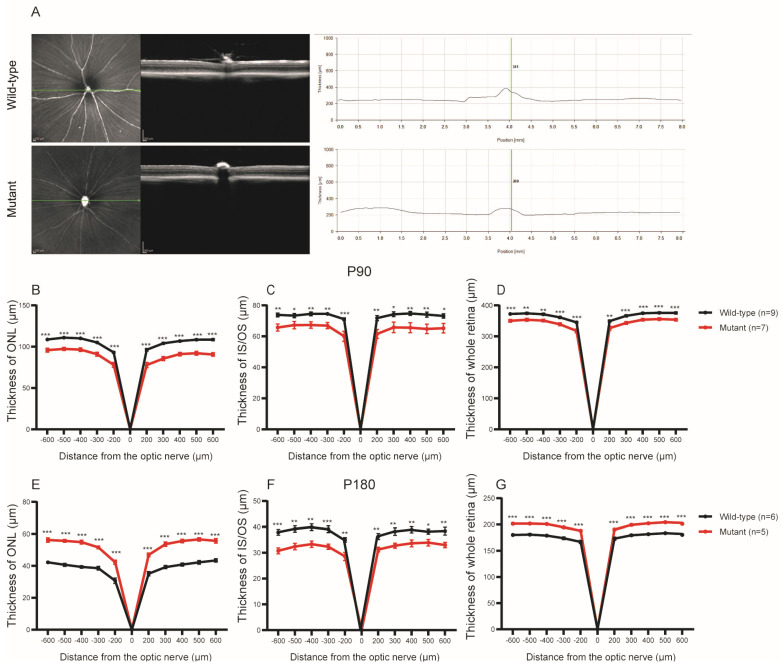
Thickness of different retinal layers of Cep250 and WT mice taken by performing spectral-domain optical coherence tomography (SD−OCT) of Cep250 and WT mice. (**A**) Representative OCT images and thickness of the whole retina of Cep250 and WT mice at P180. (**B**–**D**) Measurement of the thickness of the ONL, IS/OS, and whole retina at P90. (**E**–**G**) Measurement of the thickness of the ONL, IS/OS, and whole retina at P180. The values are presented as the means ± SEMs. * *p* < 0.05, ** *p* < 0.01, *** *p* < 0.001 (Student’s *t*-test). n: the number of animals used for statistical analysis.

**Figure 2 ijms-24-08843-f002:**
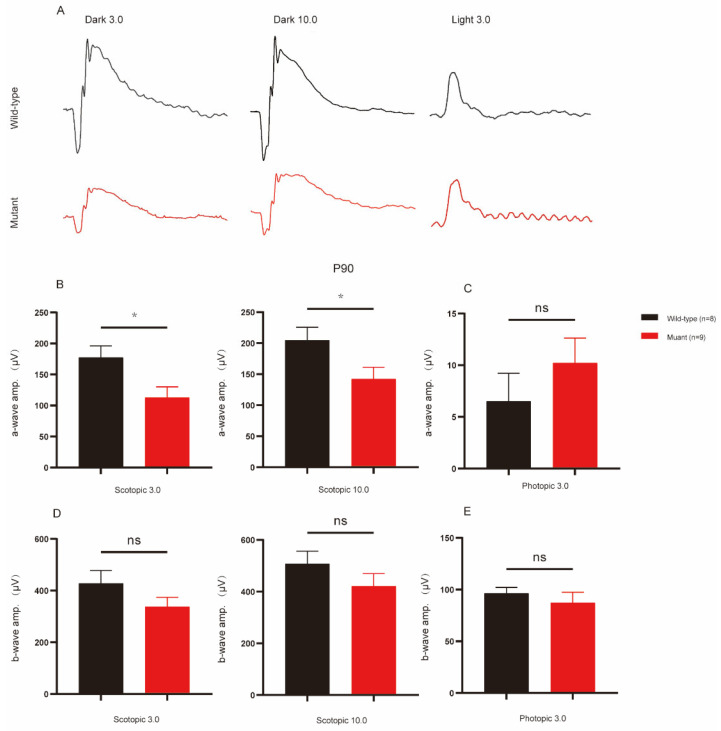
Retinal function of Cep250 and WT mice at P90 as determined by the ERG. (**A**) Representative waves under dark 3.0, dark 10.0, and light 3.0 conditions. (**B**–**E**) Statistical analysis of a-wave and b-wave amplitudes at P90. Means ± SEMs. * *p* < 0.05; ns = no significance; n: the number of animals used for statistical analysis. Student’s *t*-test.

**Figure 3 ijms-24-08843-f003:**
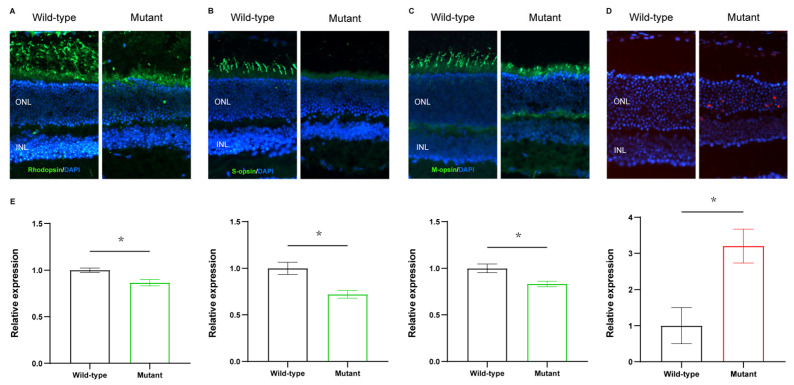
Immunostaining of photoreceptors and TUNEL staining of retinas. (**A**) Immunostaining of rhodopsin of Cep250 and WT mice at P180 (magnification: 200×). (**B**,**C**) Immunostaining of S-opsin and M-opsin of Cep250 and WT mice at P180 (magnification: 200×). (**D**) TUNEL staining of Cep250 and WT mice at P180 (magnification: 200×). (**E**) Corresponding quantitative data of the relative expression in (**A**–**D**). The values are presented as the means ± SEMs. * *p* < 0.05. Student’s *t*-test. *n* = 3.

**Figure 4 ijms-24-08843-f004:**
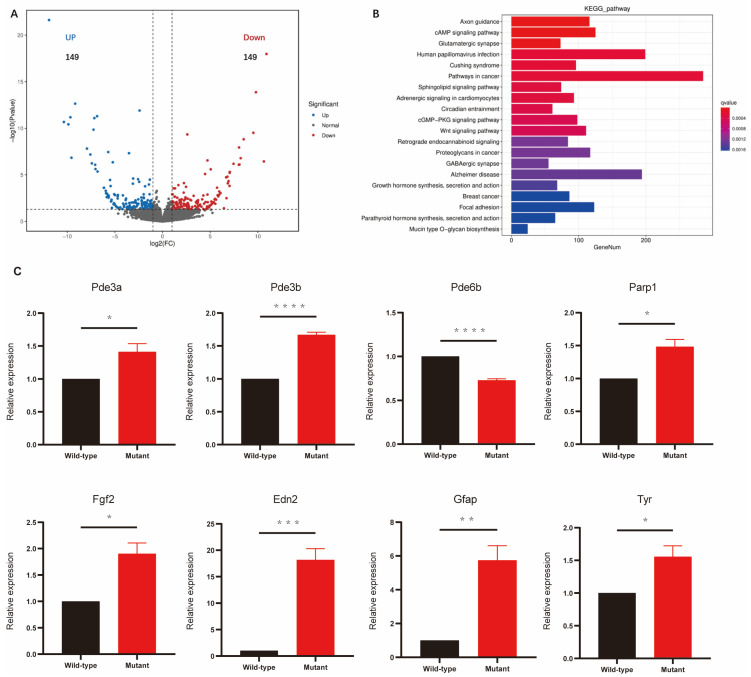
Transcriptional profiling of retinas from Cep250 KO mice and verification of gene expression. (**A**) Volcano plot showing highlights of DEGs from Cep250 KO retinas compared with WT retina. (**B**) The significantly enriched pathways in the Cep250 KO retina (compared with WT retina) from the KEGG analysis of the DEGs. (**C**) Bar figures showing the relative gene expression measured by the qPCR analysis. Means ± SEMs. * *p* < 0.05, ** *p* < 0.01, *** *p* < 0.001, **** *p* < 0.0001. Student’s *t*-test. *n* = 3.

**Table 1 ijms-24-08843-t001:** Top 15 upregulated and downregulated genes in the Cep250 KO retina compared with the WT retina.

Top 15 Upregulated Genes				Top 15 Downregulated Genes			
Symbol	Log2(FC)	*p*-Value	Description	Symbol	Log2(FC)	*p*-Value	Description
Edn2	3.905	8.609 × 10^28^	endothelin-2 preproprotein	Cryaa	−6.122	0.015	alpha-crystallin A
Lad1	2.757	3.49 × 10^14^	ladinin-1	Crygs	−5.453	0.048	gamma-crystallin S
Cyp4f15	2.184	0.003	cytochrome P450	Cryba1	−5.411	0.028	beta-crystallin A1
Dio3	2.136	0.001	thyroxine 5-deiodinase	Crybb1	−4.558	0.012	beta-crystallin B1
C4b	1.965	0.0002	complement C4-B	Cryab	−2.889	0.037	alpha-crystallin B
Itgam	1.802	0.0008	integrin alpha-M	Nid1	−2.543	0.047	nidogen-1
Tyr	1.761	5.617 × 10^6^	tyrosinase isoform 1	Cd24a	−2.153	0.049	signal transducer CD24
Xlr3a	1.678	0.0009	mCG114264, isoform CRA_b	Zbed6	−2.081	0.026	zinc finger BED domain
A2m	1.606	1.609 × 10^11^	alpha-2-macroglobulin-P	Lamb3	−1.704	0.003	laminin subunit beta-3
Dsg1a	1.566	0.0009	desmoglein-1-alpha preproprotein	Cdh1	−1.613	0.002	cadherin-1 preproprotein
Cpt1b	1.551	5.046 × 10^5^	carnitine O-palmitoyltransferase 1	Scand1	−1.600	0.007	SCAN domain
Gfap	1.524	1.434 × 10^8^	glial fibrillary acidic protein	Rps27	−1.511	0.015	40S ribosomal
Thbs2	1.490	0.025	thrombospondin-2	Lrrc3b	−1.321	0.010	leucine-rich repeat
Zic4	1.484	0.017	zinc finger protein 4	Hba-a1	−1.206	0.006	hemoglobin alpha
Cldn1	1.483	3.946 × 10^5^	claudin-1	Lgals3	−1.203	0.025	galectin-3

## Data Availability

The data presented in this study are all contained within the main body and Appendix A of this article.

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
