# Peer review of "RNA-Seq Analysis Reveals an Essential Role of the cGMP-PKG-MAPK Pathways in Retinal Degeneration Caused by Cep250 Deficiency"

_ijms, 2023, doi:10.3390/ijms24108843_

Round 1

Reviewer 1 Report

Cheng Chon et al. described in their manuscript the effects of Cep250-deficiency regarding retinal degeneration using Cep-250 deficient mice. They analysed the pathogenesis of the retinal phenotype and studied the involved signaling pathways by RNA seq analysis. Thereby, they found that several kinase pathways were upregulated including expression of cGMP/PKG signaling genes, whereas expression of protein processing genes in the endoplasmic reticulum were reduced.

The analysis is convincingly described and raises new aspects regarding the effects of Cep-deficiency in the retinal degeneration and the involved signaling pathways e.g cGMP-PKG-MAPK. There are some aspects which should be more clearly explained and should be sufficiently answered.

Major points:

-       Deletion of Cep250 gene was performed by CRISPR/CAS, however figures proving the deletion are lacking in the manuscript. Hence, the authors should show the procedure detecting the gene deletion e.g. a figure of the PCR analysis of WT in comparison to Cep250-knockout in the supplement. Furthermore, the lacking expression of the Cep250 gene in the Cep250-deficient mice should be shown e.g. in a further figure with the RT-PCR analysis in the supplement.

-       Fig. 1 and Fig. 2: Does the “n”-number in the figure indicate the number of used animals and/or of the analysed SD-OCT or ERG measurements taken for the statistics. This should be stated in the figure legends.

-       Figure 3 (page 5): magnification of the images should be stated in the figure legend. Please indicate how many wild-type and Cep250-knockout animals were analysed. Statistical analysis of the images should be given.

-       Figure 3 (page 6): should be renamed to “Figure 4”. Please indicate in this figure the P value of asterisks (what statistic value indicate **** ? This value should be also described in 4.8 Data analysis))

-       Discussion: A recent manuscript by Abu-Diab et al. (reference 8) describes another homozygous knockout of Cep250 with a relatively-late onset of retinal degeneration in which P180 did not reveal a reduced ONL thickness and ERG response, whereas in P360 these parameters were diminished. In the current manuscript by Cheng Chon et al. this late onset was not observed. Hence, these differences in comparison to the results of Abu-Diab et al. should be discussed.

-        

Minor points:

-       In the figures, there are different description for the Cep250 KO mice (e.g. in Fig. 1A: mutant; Fig. 1B-G: CEP250; Fig. 2A: Mutant; Fig. 2B-E: Hom;…), It would be helpful if you use always the same nomenclature of description for the homozygous Cep250 KO mice e.g. CEP250 and define this nomenclature in the Material and Methods section e.g. in 4.1. Animals.

Line 105: please change “th” 

Reviewer 2 Report

Dear Authors,

I have read your manuscript Chen et al. RNA-Seq Analysis Reveals an Essential Role of the cGMP-PKG-MAPK Pathways in Retinal Degeneration Caused by Cep250 Deficiency. The title of the article accurately reflects the content and main conclusion. Abstract is fully consistent with all sections of the manuscript. The manuscript itself is concise and written clearly and understandably. Using CRISPR/Cas9 system the authors generated Cep250 knockout mice modelling Usher syndrome. OCT, ERG, Tunel and s-, m- opsin immunohystochemistry were applied at p90 and p180. The thickness of ONL, IS/OS and the whole retina of CEP250 mice was significantly reduced over time. The a- wave and b-wave amplitude of Cep250 mice in scotopic and photopic ERG were lower. In the RNA-seq, a fold change (FC) of at least two and a Pvalue < 0.05 are the criteria for defining differentially expressed genes (DEGs). Eight genes were selected and validated using qRT-PCR.

According to KEGG enrichment analysis DEGs are involved in the cGMP-PKG signalling pathway, MAPK signalling pathway, insulin secretion, edn2-fgf2 axis pathways and thyroid hormone synthesis.

Minor:

Please insert bars in IHC figures.

Please add n of animals  at each figure and SD or SE at fig 3c.

Edn2-fgf2 axis pathways are not mentioned in abstract.

It's not for me to judge English, but it seems that editing is required in places of the passive voice.

Round 2

Reviewer 1 Report

Suggestions by the reviewer were sufficiently answered.

Minor corrections:

1) Figure 4: What statistic value in panel c indicates **** (Pde3b, PDE6b)? This value should be also described in 4.8 Data analysis.

2) Please change in Figure Legend 3:

"Corresponding" instead of "Corresbonding" 
